# Social Isolation and Loneliness during COVID-19 Lockdown: Associations with Depressive Symptoms in the German Old-Age Population

**DOI:** 10.3390/ijerph18073615

**Published:** 2021-03-31

**Authors:** Felix Müller, Susanne Röhr, Ulrich Reininghaus, Steffi G. Riedel-Heller

**Affiliations:** 1Institute of Social Medicine, Occupational Health and Public Health (ISAP), Medical Faculty, University of Leipzig, 04103 Leipzig, Germany; susanne.roehr@medizin.uni-leipzig.de (S.R.); steffi.riedel-heller@medizin.uni-leipzig.de (S.G.R.-H.); 2Global Brain Health Institute (GBHI), Trinity College Dublin, D02 PN40 Dublin, Ireland; 3Department of Public Mental Health, Central Institute of Mental Health, Medical Faculty Mannheim, Heidelberg University, 68159 Mannheim, Germany; ulrich.reininghaus@zi-mannheim.de; 4Centre for Epidemiology and Public Health, Health Service and Population Research Department, Institute of Psychiatry, Psychology & Neuroscience, King’s College London, London WC2R 2LS, UK; 5ESRC Centre for Society and Mental Health, King’s College London, London WC2R 2LS, UK

**Keywords:** COVID-19, lockdown, social isolation, loneliness, mental health, depressive symptoms, resilience, old age, survey

## Abstract

Lockdowns during the COVID-19 pandemic increase the risk of social isolation and loneliness, which may affect mental wellbeing. Therefore, we aimed to investigate associations between social isolation and loneliness with depressive symptoms in the German old-age population during the first COVID-19 lockdown. A representative sample of randomly selected individuals at least 65 years old (*n* = 1005) participated in a computer-assisted standardized telephone interview in April 2020. Sociodemographic data, aspects of the personal life situation, attitudes towards COVID-19 and standardized screening measures on loneliness (UCLA 3-item loneliness scale), depression (Brief Symptom Inventory/BSI-18), and resilience (Brief Resilience Scale/BRS) were assessed. Associations were inspected using multivariate regression models. Being lonely, but not isolated (β = 0.276; *p* < 0.001) and being both isolated and lonely (β = 0.136; *p* < 0.001) were associated with higher depressive symptoms. Being isolated, but not lonely was not associated with depressive symptoms. Thus, the subjective emotional evaluation, i.e., feeling lonely, of the social situation during lockdown seems more relevant than the objective state, i.e., being isolated. Normal (β = −0.203; *p* < 0.001) and high resilience (β = −0.308; *p* < 0.001) were associated with lower depressive symptoms across groups. Therefore, strengthening coping skills may be a support strategy during lockdowns, especially for lonely older individuals.

## 1. Introduction

Since early 2020, most of the world’s population has been affected by the spread of newly emerged coronavirus Severe Acute Respiratory Syndrome Coronavirus 2 (SARS-CoV-2) [1]. Germany was among the early countries with a rapidly increasing number of infections, which made the first lockdown from 22 March 2020 necessary [2]. From this day, various infection control measures in different intensities have continuously been in place. In the governmental infection control concept, social distancing plays a major role. From November 2020 on, Germany experienced the second wave of the SARS-CoV-2 pandemic with a total number of more than 2,100.000 infections and 92,457 detected infections over a period of seven days [3].

The World Health Organization (WHO) issued an early statement on mental and psychosocial health considerations during the COVID-19 outbreak [4], raising awareness on possible mental health consequences of mass quarantine measures. The concerns drew on research results from the time of previous coronavirus outbreaks, e.g., SARS-CoV in 2002/2003, demonstrating an association between quarantine measures and negative psychosocial health consequences including depressive symptoms, anxiety, anger, and stress [5]. 

Early in the pandemic, the older individuals were identified as a risk group for mental health impacts, because older age was quickly established as the main risk factor for severe and fatal courses of COVID-19 [6]. Potentially, the resulting fear of the virus and the associated recommendation for particularly strict social distancing (“cocooning”) were suspected of having a negative impact on mental health. The British Royal College of Psychiatrists, therefore, presumed the need for mental health support of older people likely to be greater than ever [7]. However, initial studies on the psychological impact of the pandemic did not confirm older individuals being a vulnerable group with regards to mental health. First analyses of data used in this study showed that the mental wellbeing in the German old age population was largely unaltered during the early pandemic [8]. Mental health outcomes, including depression, anxiety, somatization, and overall psychological distress were compared with data from earlier representative cross-sectional studies in the old-age population, with no significant differences being found. These conclusions are consistent with results from two longitudinal studies comparing data of the UK and US population before and during the pandemic, finding just a slight, but not clinically relevant change in psychological distress. Rather, both studies showed a strong age gradient with younger people being more psychologically distressed [9,10].

Despite the fact that the mental health of the general German older population seemed to be largely unaltered in the short term, there may be certain risk groups who may have been impacted during the pandemic [8]. Two of them could be individuals who are isolated and/or lonely, which is a likely result of the infection control measures, especially lockdowns. An association between loneliness and poor mental health is well established. A focus of many studies was the relationship with depressive symptoms [11]. For social isolation, the evidence is less clear. However, there are some studies suggesting an association between social isolation and depressive symptoms [12,13]. In addition to the effect on mental health, previous studies have also shown a relationship between social isolation and loneliness with increased mortality [14], decreases in cognitive functioning [15], and increased risk of Alzheimer’s Disease [16]. This reinforces the importance of studying social isolation and loneliness in the old age population. 

Reviewing the evidence between social isolation and loneliness and mental health on older individuals, Courtin and Knapp found consistent associations with depressive symptoms [17]. However, the authors noted, on the one hand, there were significantly fewer studies that focused on social isolation compared to loneliness and, on the other hand, there was a huge variety of measurements for social isolation. This was interpreted as the main reason why the association with depressive symptoms was usually found to be weaker than for loneliness. Only a few studies examined the association of both social isolation and loneliness with depressive symptoms in older individuals. 

Another factor that may be associated with depressive symptoms as a result of the COVID-19 pandemic is resilience, i.e., the capability to adapt to and recover from stressful events. Previous studies showed that older individuals were less prone to adverse mental health outcomes related to the COVID-19 pandemic than younger individuals [8,9,10]. One reason for this could be higher resilience. Resilient individuals may cope better with the pandemic situation which may result in lower depressive symptoms.

As a result of the restrictions to curb the COVID-19 pandemic, we are facing a situation of increased risk of social isolation and loneliness, deeming it timely and relevant to inspect associations with mental health. Individuals, who were previously well integrated, may now face isolation from close ones without the possibility of direct contact. It is also conceivable that, due to social distancing, feelings of loneliness arose in individuals who previously did not experience them. The pandemic situation may be particularly challenging for individuals who may have had small social networks before the pandemic already. Especially older individuals live alone more often than younger individuals. 

Against this background, we aimed to investigate the association between social isolation and loneliness with depressive symptoms and the effects of resilience in the old age population during the first COVID-19 lockdown in Germany.

## 2. Materials and Methods

### 2.1. Study Population

The sample is part of a cross-sectional study conducted by USUMA, a leading social research institute in Germany. Being at least 65 years of age was the only criteria to be included in the study. Furthermore, participation required informed consent, which was agreed upon verbally at the beginning of the telephone interview. The target sample size was 1000 individuals, who were invited to participate in a computer-assisted telephone interview. Therefore, the research institute had to contact 1863 individuals (53.68%). Participants were selected using random digit dialing, drawing from the Association of German Market and Social Research Agency’s (ADM) telephone number sample base. Random selection of households was guaranteed by drawing telephone numbers proportionally to the German population structure and regional stratification according to district sizes across Germany. The Kish-Selection grid [18] was used to randomly choose a person to participate in the study if there was more than one person older than 65 years living in the same household. Individuals of the target group were called over the phone by trained interviewers. Parallel to the telephone interview, data was recorded using a computer-based data collection mask. To ensure representativeness, a weight variable was calculated to account for sample deviations from the target population with regards to age, sex, and regions across Germany, using official population statistics by the Federal Statistical Office from the year 2019. Data were collected from 6 April to 25 April 2020, when the first nationwide COVID-19 lockdown was continuously in force.

### 2.2. Measurements

Telephone interviews were structured in three parts. First, participants were asked to provide a range of sociodemographic data. This included standardized questions on age (years), sex (female/male/other), education (low/middle/high), marital status (married, single, divorced, widowed), and living situation (alone, with partner/spouse, with relatives others than partner/spouse, with others).

Second, variables related to the COVID-19 pandemic were surveyed, comprising attitudes to and compliance with mass quarantine measures and aspects of the personal life situation. Attitudes to and compliance with mass quarantine measures were assessed using 5-point Likert-scales. This included worry due to the COVID-19 pandemic, perceived threat by COVID-19, the supportiveness of the governmental infection control measures, and the subjectively felt restriction due to governmental measures. Regarding the personal life situation, the frequency of direct contacts with individuals outside of the own household over the past week (“no contact at all” to “several times a day”) and the duration of quarantine measures (days between 22nd of March, when quarantine measures in Germany started, and time of the interview) were assessed.

As a third part of the interview, standardized screening measures were used to examine psychosocial health outcomes, i.e., loneliness, resilience, and depressive symptoms. 

Loneliness was assessed using the 3-item version of the University of California, Los Angeles Loneliness Scale (UCLA-3) [19]. UCLA-3 consists of 3 items evaluating the subjectively perceived loneliness of participants. Possible answers range from “never” to “often” (scored 0 to 3). After calculating the sum score, a cut-off score of ≥6 indicated loneliness. This cut-off score was used in many previous studies [20,21]. The UCLA-3 is often used in telephone interviews as a reliable and valid measure for loneliness [22].

As an indicator for social isolation, we used information on household composition and frequency of direct contact with others. Therefore, we considered everyone being socially isolated, who (a) lived alone and (b) had no direct contacts over the past week. We combined the information on social isolation and loneliness into one variable with four possible states: 1—not isolated and not lonely, 2—isolated and not lonely, 3—not isolated and lonely, 4—isolated and lonely.

To measure resilience, we applied the validated German version [23] of the Brief Resilience Scale (BRS) [24]. The scale consists of 6 items assessing the ability to recover from stress on a 5-point Likert-scale ranging from “strongly disagree” to “strongly agree”. Three items were negatively worded to reduce response bias in relation to social desirability and, therefore, had to be coded in reverse. To quantify resilience the mean score of all item responses was used (range 1-5). Higher scores indicated higher resilience. It was classified as 1.00–2.99 = low resilience, 3.00–4.30 = normal resilience, 4.31–5.00 = high resilience [24].

To measure depressive symptoms, we used the depression scale of the Brief Symptom Inventory (BSI-18). The BSI-18 consists of 18 items assessing depression, anxiety, and somatization with 6 questions each. The frequency of depressive symptoms in the past week was assessed on a 5-point Likert-scale of 0 (“not at all”) to 4 (“very much”). Thus, the sum scores of the BSI-18 depression subscale had a total range from 0 to 24. The scale was used as an outcome for linear regression models; therefore, no cut-off score was applied. For the German version of the BSI-18, similarly good psychometric qualities as for the American original have been reported [25].

### 2.3. Statistical Analysis

One-way ANOVA or χ2 test were applied to detect differences across groups of loneliness and social isolation in sociodemographic characteristics, duration of quarantine, attitudes towards COVID-19, resilience, and BSI-18 depression sum score. Furthermore, three multivariate regression models were composed to examine associations between social isolation and loneliness with depressive symptoms. The continuous sum score of the BSI-18 depression scale was used as an outcome for all regression models. First, an unadjusted model with social isolation and loneliness as an independent categorical variable was conducted (categorized into 4 groups: isolated and not lonely, not isolated and lonely, isolated and lonely in reference to not isolated, and not lonely). In the second model, we adjusted for age (continuous), gender (dichotomous; female in reference to male), education (categorical; categorized according to the Comparative Analysis of Social Mobility in Industrial Nations/CASMIN classification; low, middle in reference to high) [26] and marital status (categorical; single, divorced and widowed in reference to married). The final model included attitudes towards COVID-19 (continuous), duration since lockdown (continuous; in days), and resilience (categorical; normal and high in reference to low). Standardized beta (β) coefficients are reported. All analyses applied sampling weights and were performed using SPSS Statistics 25.0. The level of significance was set to *p* < 0.05. 

## 3. Results

### 3.1. Descriptive Analysis

Table 1 and Table 2 summarize descriptive data for all variables of interest for the total sample of participants as well as for social isolation and loneliness subgroups. The mean age of the participants was 75.5 years (SD = 7.1; range = 65–94) with 56.3% being female. Participants were distributed among the 4 groups formed for social isolation and loneliness as follows: 76.1% not isolated and not lonely, 10.9% isolated and not lonely, 10.6% not isolated and lonely, 2.5% isolated and lonely.

Between the four subgroups, there were significant differences in the following variables: age, gender, education, marital status, worry due to COVID-19, perceived threat by COVID-19, feeling restricted by quarantine measures, depression score, and resilience score. Participants, that were isolated but not lonely were considerably older than participants of the other three subgroups (M = 78.94, SD = 8.01 vs. M = 75.08, SD = 6.8; M = 75.02, SD = 7.54; M = 75.18, SD = 7.11). Women were more often socially isolated or lonely than men (12.9%; 12.9%; 3.4% vs. 8.3%; 7.6%; 1.2%). Married participants experienced social isolation or loneliness less often than individuals who were not married (90.1% vs. 58.4%; 59.0%; 57.4%). Participants who were strongly worried about COVID-19 were more frequently not isolated and lonely or isolated and lonely than participants who were not worried at all (12.8% and 3.3% vs. 1.9% and 1.9%). The same applied for perceived threat by COVID-19 (14.3% and 3.9% vs. 7.7% and 1.4%). In line with this, the proportion of those being not isolated and lonely or isolated and lonely among participants who felt severely restricted by quarantine measures was significantly higher than among those who did not feel restricted at all (11.3% and 6.6% vs. 3.5% and 1.5%). The resilience score was slightly higher in not lonely than in lonely participants (3.61 and 3.59 vs. 3.4 and 3.3). Matching this, the depression score was considerably higher in lonely than in not lonely participants (3.28 and 3.47 vs. 1.08 and 1.26). For the duration of lockdown and being supportive of quarantine measures, no differences between subgroups were detected. Further results on study sample characteristics are presented in Table 1 and Table 2.

### 3.2. Linear Regression Analysis

Table 3 shows associations between social isolation and loneliness with depressive symptoms. Being not isolated and lonely and being isolated and lonely were both significantly associated with higher depression scores. This association remained significant after adjusting for confounding variables (both *p* < 0.001). Notably, the impact of being not isolated and lonely on the depression score was even higher than it was for being isolated and lonely (β = 0.276 vs. β = 0.136). Accordingly, no association between being isolated and not lonely and depressive symptoms was found. Among the other predictor variables, being single (β = 0.113; *p* < 0.001), being divorced (β = 0.059; *p* = 0.046), being widowed (β = 0.191; *p* < 0.001), being worried due to COVID-19 (β = 0.089; *p* = 0.005) and feeling restricted by quarantine measures (β = 0.083; *p* = 0.004) were associated with higher depression scores. In contrast to that, being supportive of quarantine measures (β = −0.061; *p* = 0.035) and normal (β = −0.203; *p* < 0.001) and high resilience (β = −0.308; *p* < 0.001) were associated with lower depressive symptoms.

## 4. Discussion

We investigated the association of social isolation and loneliness with depressive symptoms in the old-age population (≥65 years) during the first COVID-19 lockdown in Germany. Our findings concern experiences after the initial social distancing measures were in force for an average of 28 days. Loneliness was strongly associated with depressive symptoms. This is consistent with previous studies prior to and during the COVID-19 pandemic [11,27,28,29,30,31]. For example, Lee et al. demonstrated an association of loneliness and depressive symptoms with data from The English Longitudinal Study of Ageing (ELSA) for the population aged 50 years and older [28]. First studies conducted during the early COVID-19 pandemic also showed an association between loneliness and depressive symptoms [29,30,31]. However, these studies mostly provided data for the general population. Studies that examine this association in the old age population during the pandemic are still scarce. Our study was, therefore, able to contribute to this need for research by showing a significant association of loneliness and depressive symptoms in the German old age population during the time of the COVID-19 pandemic. 

Our data did not indicate an association between social isolation and depressive symptoms. This relationship has been investigated by significantly fewer studies. Accordingly, the evidence is less clear. Hämmig showed an association between social isolation and moderate to severe depression in different age groups [12]. Another study showed a relation between social isolation in terms of a weak connectedness with relatives and with friends and depressive symptoms. Other assessed aspects of social isolation were not associated with depressive symptoms [13]. In the 2012 publication by Coyle and Dugan, no association was found between social isolation and having a mental health problem in adults aged 50 years and older [32]. However, some aspects may limit the comparability of study results: On the one hand, different measures were used for the assessment of social isolation. Some studies used self-created indicators [12,32], others different validated measuring scales [13]. A separation between the constructs of loneliness, as a subjective feeling, and social isolation, as an objective indicator, is not always given [12]. A clear conclusion in comparison to the existing evidence is therefore only possible to a limited extent. However, the strength of our way of assessing social isolation is that it is clearly differentiated from loneliness and is very strictly defined with (a) living alone and (b) having no direct contact per week. Overall, our results indicate that the subjective emotional evaluation, i.e., of feeling lonely, of the social situation during lockdown seems more relevant than the objective state, i.e., of being isolated. Thus, especially individuals reporting loneliness in lockdown scenarios should be targeted for public health interventions.

The vast majority of our participants who lived alone reported having strictly followed social distancing recommendations. For this group, no association of social distancing measures and depressive symptoms was found. Results from a study from Hong Kong suggested that compliance with social distancing measures can have a positive effect on depressive symptoms [33]. However, such results represent the status during the early phase of the pandemic. How this relationship presents over time after repeated lockdowns must be investigated with longitudinal data.

Additionally, it is noteworthy to discuss the association between resilience and lower depressive symptoms. Resilience was the strongest of all covariates examined. This illustrates the importance of personal capacities for coping with crises such as the COVID-19 pandemic. As noted by Kimhi et al., individual resilience is a much better predictor of coping with the consequences of the pandemic than national and community resilience is [34]. This, in turn, demonstrates the need to identify groups with low resilience, who may need help in coping with the situation to attenuate or avoid adverse health outcomes.

As could be shown, there are certain risk groups, e.g., lonely older individuals, for adverse mental health outcomes during the COVID-19 pandemic. These individuals should be provided support and awareness from the public health perspective. Digital solutions could be particularly useful tools in this regard. [35,36]. However, barriers to implementing telehealth approaches in the old age population need to be considered, for example, access to the internet, digital literacy, and attitudes towards digital technologies [35]. On the other hand, the increased need for social connectedness under pandemic circumstances may accelerate their usage and acceptance in the old age population. Both self-guided and clinician-guided interventions should be considered. Tomasino and colleagues found peer-supported internet interventions to be equivalent to expert-delivered internet programs. Therefore, including peer support may be a potentially more cost-effective way for delivering online treatments for depression, which also may reduce clinician burden and increase social interaction [35,37]. Low-threshold programs such as educational outreach or wellness guides could also help to reduce depressive symptoms [38]. According to a survey among members of the German Psychotherapists Association, the demand for psychotherapeutic treatment increased due to the COVID-19 pandemic. Psychotherapists received 40.8% more inquiries in January 2021 than in January 2020, which also increased the waiting time for an initial consultation [39]. Therefore, telehealth programs could also help to bridge the treatment gap.

### Strengths and Limitations

One of the strengths of the study is the representative design. The data collection took place in April 2020, shortly after the introduction of the first lockdown in Germany, which allowed conclusions to be drawn about the immediate impact of the pandemic. We were able to investigate the associations between both social isolation and loneliness with depressive symptoms. This sets it apart from other studies with mostly one of the factors only. The study focused on the older population, a particular risk group in the COVID-19 pandemic, which, in contrast to many other studies focusing on the general population, allowed for more specific and well-founded conclusions for this age group. 

A limitation of the study is the cross-sectional design, which only provided a snapshot of the situation during the initial phase of the pandemic. Therefore, research over the course of the pandemic from a longitudinal perspective is needed. Due to different definitions and measurements of social isolation, the comparison with other studies is limited. In addition, other factors that might influence the associations of interest, such as pre-existing depressive symptoms or other physical and psychiatric conditions, could not be considered. As this is a cross-sectional study, we cannot rule out, that such pre-existing symptoms and conditions may be associated with responses to the COVID-19 pandemic.

## 5. Conclusions

Our results showed that lonely older individuals were at risk for higher depressive symptoms during the first COVID-19 lockdown in Germany. Social isolation, on the other hand, was not associated with depressive symptoms. This indicates that the subjective emotional evaluation, i.e., feeling lonely, of the social situation during lockdown matters more for mental wellbeing than the objective state, i.e., being isolated, for a period of time. As higher resilience was strongly associated with lower depressive symptoms, strengthening coping skills to better endure the pandemic may be a useful support strategy with regards to mental health. This may be especially relevant for lonely older individuals—as the pandemic is lasting and repeated lockdowns are enforced. As our study only represents the situation during the initial phase of the pandemic, future investigations should examine associations over the course of the pandemic as well as post-pandemic from a longitudinal perspective. Identifying further risk groups for adverse mental health outcomes during the COVID-19 pandemic is necessary in order to provide suitable public health interventions.

## Figures and Tables

**Table 1 ijerph-18-03615-t001:** Sociodemographic characteristics by social isolation/loneliness subgroups.

	Total	Not Isolated and Not Lonely	Isolated and Not Lonely	Not Isolated and Lonely	Isolated and Lonely	Group Difference (*p*-Value)
Total; *n* (%)	993	755 (76.1)	108 (10.9)	105 (10.6)	25 (2.5)	
Sex; *n* (%)						
Female	559	396 (70.8)	72 (12.9)	72 (12.9)	19 (3.4)	<0.001
Male	433	359 (82.9)	36 (8.3)	33 (7.6)	5 (1.2)	
Age; M (SD)	75.5 (7.1)	75.08 (6.8)	78.94 (8.01)	75.02 (7.54)	75.18 (7.11)	<0.001
Education; *n* (%)						
Low	276	204 (73.9)	42 (15.2)	24 (8.7)	6 (2.2)	0.002
Middle	347	267 (76.9)	39 (11.2)	28 (8.1)	13 (3.7)	
High	355	274 (77.2)	26 (7.3)	50 (14.1)	5 (1.4)	
Marital status; n (%)						
Married	555	500 (90.1)	6 (1.1)	48 (8.6)	1 (0.2)	<0.001
Single	77	45 (58.4)	18 (23.4)	11 (14.3)	3 (3.9)	
Divorced	100	59 (59.0)	18 (18.0)	11 (11.0)	12 (12.0)	
Widowed	258	148 (57.4)	67 (26.0)	34 (13.2)	9 (3.5)	

Missing values: Social Isolation/Loneliness: *n* = 12 (1.2%); Education: *n* = 13 (1.3%); Marital Status: *n* = 4 (0.4%).

**Table 2 ijerph-18-03615-t002:** Attitudes towards COVID-19 and resilience and depression scores by social isolation/loneliness subgroups.

	Total	Not Isolated and Not Lonely	Isolated and Not Lonely	Not Isolated and Lonely	Isolated and Lonely	Group Difference (*p*-Value)
Being worried about COVID-19; *n* (%)						
Totally disagree	107	87 (81.3)	16 (15.0)	2 (1.9)	2 (1.9)	0.002
Disagree	128	90 (70.3)	24 (18.8)	12 (9.4)	2 (1.6)	
Neutral	226	172 (76.1)	26 (11.5)	21 (9.3)	7 (3.1)	
Agree	173	138 (79.8)	10 (5.8)	24 (13.9)	1 (0.6)	
Totally agree	359	268 (74.7)	33 (9.2)	46 (12.8)	12 (3.3)	
Perceived threat by COVID-19; *n* (%)						
Totally disagree	143	108 (75.5)	22 (15.4)	11 (7.7)	2 (1.4)	0.011
Disagree	207	146 (70.5)	32 (15.5)	26 (12.6)	3 (1.4)	
Neutral	286	230 (80.4)	26 (9.1)	23 (8.0)	7 (2.4)	
Agree	125	105 (84.0)	6 (4.8)	11 (8.8)	3 (2.4)	
Totally agree	230	166 (72.2)	22 (9.6)	33 (14.3)	9 (3.9)	
Being supportive of quarantine measures; *n* (%)						
Totally disagree	7	5 (71.4)	2 (28.6)	0 (0)	0 (0)	0.703
Disagree	9	6 (66.7)	1 (11.1)	1 (11.1)	1 (11.1)	
Neutral	79	60 (75.9)	7 (8.9)	9 (11.4)	3 (3.8)	
Agree	117	90 (76.9)	9 (7.7)	15 (12.8)	3 (2.6)	
Totally agree	778	591 (76.0)	90 (11.6)	80 (10.3)	17 (2.2)	
Feeling restricted by quarantine measures; *n* (%)						
Totally disagree	198	155 (78.3)	33 (16.7)	7 (3.5)	3 (1.5)	<0.001
Disagree	213	175 (82.2)	22 (10.3)	14 (6.6)	2 (0.9)	
Neutral	308	234 (76.0)	18 (5.8)	48 (15.6)	8 (2.6)	
Agree	123	88 (71.5)	12 (9.8)	20 (16.3)	3 (2.4)	
Totally agree	151	102 (67.5)	22 (14.6)	17 (11.3)	10 (6.6)	
Duration since lockdown; M (SD)	27.98 (4.76)	27.97 (4.69)	28.36 (5.05)	27.31 (5.17)	29.58 (3.15)	0.135
Resilience						
M (SD)	3.58 (0.67)	3.61 (.68)	3.59 (0.65)	3.40 (0.61)	3.30 (0.77)	0.004
High; *n* (%)	131	89 (67.9)	15 (11.5)	20 (15.3)	7 (5.3)	
Normal; *n* (%)	639	487 (76.2)	68 (10.6)	71 (11.1)	13 (2.0)	
Low; *n* (%)	174	140 (80.5)	21 (12.1)	100 (10.6)	4 (2.3)	0.027
Depression; M (SD)	1.38 (1.98)	1.08 (1.61)	1.26 (1.71)	3.28 (2.9)	3.47 (2.9)	<0.001

Abbreviations: M Mean; SD standard deviation; *p p*-value; Missing values: Being worried about COVID-19: *n* = 1 (0.1%); Perceived threat by COVID-19: *n* = 1 (0.1%); Being supportive of quarantine measures: *n* = 3 (0.3%); Feeling restricted by quarantine measures: *n* = 2 (0.2%); Resilience: *n* = 50 (4.8%); Depression: *n* = 11 (1.1%).

**Table 3 ijerph-18-03615-t003:** Results of multiple regression analyses: associations of social isolation, loneliness, sociodemographic factors, attitudes towards COVID-19, and resilience with depressive symptoms.

	Model 1			Model 2			Model 3		
	β	SE	*p*	β	SE	*p*	β	SE	*p*
Social Isolation/Loneliness (ref. not isolated and not lonely)									
Isolated and not lonely	0.029	0.191	0.336	−0.059	0.203	0.062	−0.040	0.195	0.182
Not isolated and lonely	0.338	0.193	<0.001	0.313	0.192	<0.001	0.276	0.185	<0.001
Isolated and lonely	0.186	0.380	<0.001	0.159	0.383	<0.001	0.136	0.386	<0.001
Female gender (ref. male)				<0.001	0.123	0.994	−0.013	0.118	0.655
Age				0.088	0.009	0.004	0.040	0.008	0.183
Education (ref. high)									
Low				0.020	0.608	0.543	−0.009	0.140	0.771
Middle				−0.020	−0.600	0.549	−0.025	0.132	0.442
Marital Status (ref. married)									
Single				0.127	0.225	<0.001	0.113	0.216	<0.001
Divorced				0.066	0.206	0.036	0.059	0.197	0.046
Widowed				0.171	0.157	<0.001	0.191	0.151	<0.001
Duration since lockdown							−0.013	0.012	0.639
Being worried about COVID-19							0.089	0.046	0.005
Perceived threat by COVID-19							0.011	0.045	0.720
Being supportive of quarantine measures							−0.061	0.077	0.035
Feeling restricted by quarantine measures							0.083	0.043	0.004
Resilience (ref. low)									
Normal							−0.203	0.148	<0.001
High							−0.308	0.189	<0.001
R²	0.138			0.179			0.257		

Abbreviations: β: standardized beta coefficient; SE standard error.

## Data Availability

Data are publicly available at the Figshare repository and can be accessed at https://doi.org/10.6084/m9.figshare.13013657.v1 (accessed on 3 March 2021). Data are made available and are shared under the CC BY 4.0 license.

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
