# Peer review of "Social Isolation and Loneliness during COVID-19 Lockdown: Associations with Depressive Symptoms in the German Old-Age Population"

_ijerph, 2021, doi:10.3390/ijerph18073615_

Round 1

Reviewer 1 Report

The manuscript is well-written and interesting showing many strengths such as the representative design and the size of the sample of older individuals. The topic is important and has a high relevance. However, I have some suggestions that could further improve the quality of the very interesting manuscript.

  • Maybe, the authors could also shortly introduce resilience in the introduction as it is part of the investigation – what was the rationale to investigate this construct?
  • Please avoid to repeat results in the discussion section without to embed them into the current state of research.
  • The authors could give more details on future directions – what future investigations would be desirable in this field?
  • Authors should discuss the limitations of the study in few more sentences. What sources of bias can be anticipated?
  • Additionally, the authors could expand on the given recommendations for support strategies during lockdown periods.

Reviewer 2 Report

I red with interest your paper. In my opinion it is well written and stuctured.

In particular: 1) the Introduction is interesting and relevant to the topic research. The background is well presented and the references are enough.

About the Materials and methods and Analysis I found some points not enough clear. Please, intervene on these.

  • It’s not clear in which way the computer-assisted telephone interview was conducted
  • Why the target sample was 1000 individuals? With which criteria?
  • The only inclusion criteria were age over 65?
  • The presence of previous depressive symptoms or illness regardless the pandemic could have influenced your results. How did you control them? If they have not been considered, I suggest to add them among the limitations
  • How did you check the normal distribution?

Regarding the Discussion, I appreciate the idea of implement telehealth approaches in the old age population in order to provided support and awareness about public health. Maybe could be inserted also some experiences in psychiatric unit or in other hospital units that move on in the same direction?

I suggest some minor methodological corrections before acceptance. 
